# Smoking and Diabetes Attenuate Number of CD34^+^ Haematopoietic Stem Cells in Peripheral Blood of Patients with Advanced Peripheral Artery Disease

**DOI:** 10.3390/ijms242015346

**Published:** 2023-10-19

**Authors:** Barbara Sernek, Rok Kamnikar, Miran Sebestjen, Anja Boc, Vinko Boc

**Affiliations:** 1Faculty of Medicine, University of Ljubljana, 1000 Ljubljana, Slovenia; sernek.barbara@gmail.com (B.S.); kamnikar.rok@gmail.com (R.K.); anja.boc@mf.uni-lj.si (A.B.); 2Department of Vascular Diseases, University Medical Centre Ljubljana, 1000 Ljubljana, Slovenia; vinko.boc@kclj.si; 3Department of Cardiology, University Medical Centre Ljubljana, 1000 Ljubljana, Slovenia

**Keywords:** CD34^+^ cells, peripheral artery disease, smoking, diabetes type II, intermittent claudication

## Abstract

Peripheral artery disease (PAD) is a globally prevalent problem with limited treatment options, leaving up to a fifth of patients remediless. The emergence of new studies on cell therapy in recent years offers a new promising option for their treatment. Our aim was to explore how the number of CD34^+^ hematopoietic cells in the peripheral blood of PAD patients is associated with patients’ functional as well as atherogenic factors. We selected 30 patients with advanced PAD, recorded their performance in a walking test in standard conditions and sampled their blood for further analysis with an emphasis on CD34^+^ cell selection and counting. No correlation of the CD34^+^ cell number was confirmed with any of the observed laboratory parameters. There was an association between the claudication distance and the number of CD34^+^ cells (r = −0.403, *p* = 0.046). The number of CD34^+^ cells differed between patients with and without type II diabetes (*p* = 0.071) and between active smokers, past smokers, and non-smokers (*p* = 0.035; *p* = 0.068, *p* = 0.051, respectively), with both smoking and presence of diabetes type II having a negative effect on the number of CD34^+^ cells. Our study demonstrated a dependence of the CD34^+^ cell number on the patient’s characteristics.

## 1. Introduction

Peripheral artery disease (PAD) is an increasingly prevalent manifestation of atherosclerosis in the ageing world population, exposing patients to major adverse cardiovascular events and increased all-cause mortality [1]. The current treatment measures for PAD fall into two major categories, namely best medical treatment to prevent any adverse cardiovascular event, and improvement of limb blood supply, including exercise training and revascularisation of compromised peripheral arteries in patients with disabling intermittent claudication or chronic limb-threatening ischaemia [2].

Despite the advances in revascularisation procedures, almost one-fifth of patients with advanced PAD are still not candidates for such procedures [3]. For them, a new option is emerging in the form of therapy with CD34-expressing cells gathered from bone marrow, peripheral blood, umbilical blood, or adipose tissue [4]. CD34 is a surface glycoprotein expressed in cells of various tissues, notably haematopoietic stem cells, endothelial progenitor cells, and embryonal fibroblasts. Intramuscular injections of purified CD34^+^ cells can reduce the severity of PAD in patients who are not candidates for surgical or percutaneous interventions [5].

In recent years, a variety of cell therapies have been studied as a potential treatment of ischaemia through enhancing angiogenesis. Most studies have used unselected mononuclear cells (MNC) harvested from either autologous bone marrow or peripheral blood after granulocyte-colony stimulating factor (G-CSF)-stimulated mobilisation [6]. In some of the trials, more homogenous cells have been used, including bone marrow-derived CD34^+^ cells, tissue repair monocytes (CD14^+^/CD45^+^ cells), or progenitor cells with high aldehyde dehydrogenase (ALDH) activity [7]. The first-ever trial of Therapeutic Angiogenesis by Cell Transplantation (TACT) proved the efficiency of intramuscular transplantation of autologous bone marrow-derived MNC into a swine heart [8]. Later, a clinical study suggested that the use of more homogeneous cells instead of the MNC could lead to a higher concentration of endothelial progenitor cells [5]. Autologous cell therapy using CD34^+^ cells for critical leg ischaemia (ACT34-CLI) [4] showed a borderline significant decrease in the limb amputation rate after intramuscular application of a high dose of G-CSF-mobilised CD34^+^ cells, thus indicating the number of transplanted stem cells to be one of the important factors of successful treatment. The number and functionality of CD34^+^ cells decrease with age, and their number is negatively affected by the presence of cardiovascular disease or its risk factors [7].

Our cross-sectional study aimed to explore how the clinical markers of PAD severity as well as certain inflammatory and angiogenic factors are associated with the number of CD34^+^ cells in the peripheral blood of patients with advanced PAD.

## 2. Results

### 2.1. Clinical Characteristics of Patients

The characteristics of 30 study participants are presented in Table 1. Their ages ranged from 46 to 81 years, and the majority (27/30, 90%) were older than 60 years. More than half—19 (63.3%) patients—had a clinical manifestation of atherosclerosis in other sites: 10 (33.3%) had coronary artery disease, 8 (26.7%) had carotid artery disease, and 1 (3.3%) patient had an abdominal aortic aneurysm.

### 2.2. Biochemical Characteristics of Patients

The biochemical characteristics of our patients are presented in Table 2. Their lipid values were not sufficiently regulated, given that all the patients had a secondary prevention of atherosclerosis. At the same time, moderate anaemia was present.

### 2.3. Clinical Markers of Peripheral Artery Disease Severity

Table 3 presents the values of the haemodynamic tests of the patients. Among 7 patients with an ankle–brachial index (ABI) < 0.4, only two presented with a clinical picture of chronic limb-threatening ischaemia. The remaining patients experienced intermittent claudication, with an average walking distance of 116 ± 67 m. In two patients, ABI was in the normal range (0.91–1.4), and thus not indicative of PAD. This is most likely due to mediocalcinosis, and they were included in the study based on their toe pressure.

### 2.4. Correlations of CD34^+^ Cells Number with Clinical and Biochemical Markers

We did not confirm any statistically significant correlation between the number of CD34^+^ cells and the observed clinical characteristics (Table 4) or laboratory characteristics in our patients. Among markers of PAD severity, only claudication distance was associated with the number of CD34^+^ cells (r = −0.403; *p* = 0.046) (Table 5).

Although the risk factors for atherosclerosis did not prove to be statistically significantly correlated with the number of CD34^+^ cells, there was a borderline significant difference between the patients with and without type II diabetes (Figure 1); as for smoking, there was a significant difference between the active smokers and non-smokers, and a borderline difference between the past smokers and non-smokers, and also between the active and past smokers (Figure 2).

## 3. Discussion

To our knowledge, this is the first study examining the connection between the CD34^+^ cell count and the markers of PAD severity and risk factors for atherosclerosis (diabetes mellitus, smoking, dyslipidaemia) in patients with advanced PAD.

Our results did not yield a statistically significant correlation between the number of CD34^+^ cells and age, although it has been previously proven in a study by Kresnik et al. [9]. According to their findings, the number of CD34^+^ cells declines sharply at ages above 40 years; the CD34^+^ count decreases with ageing, both under basal conditions as well as upon G-CSF stimulation. Concomitantly, age also negatively affects the function of CD34^+^ cells [9]. The probable reason for the lack of correlation in our study is the patients’ age structure, which differed considerably—the average age of our patients was approximately the same as the age of the eldest patient in the aforementioned study.

Unexpectedly, we did not find a correlation between the number of CD34^+^ cells and BMI. This result is in contrast with the findings of Müller-Ehmsen et al. [10], who reported a great reduction in CD34^+^ cell count in otherwise healthy study participants with a BMI ≥ 30 kg/m^2^, whereas the mean BMI in our patients was 26.7 kg/m^2^. In the study by Panagiotakos et al., the concentration of tumour necrosis factor α (TNFα) turned out to be proportional to BMI [11]. Therefore, a key link in the negative association between CD34^+^ cell count and obesity could be TNFα, one of the most important inhibitors of haematopoiesis in the bone marrow, both in the basal state and after G-CSF stimulation [12].

In our study, the number of CD34^+^ cells was not associated with smoking load, expressed in the number of pack-years. Contrarily, Zhen et al. [13] observed that the number of CD34^+^ cells released after G-CSF stimulation decreased with the increase in smoking load in otherwise healthy adult participants. Our findings seem to correlate with previous studies examining the effects of smoking on the number of circulating endothelial progenitor cells, including CD34^+^ cells; however, no conclusions can be drawn from our results due to non-significant *p*-values, which could be the consequence of our small sample size. Despite the lack of statistically significant differences when pack-years were considered, our active smoking participants had a significantly lower number of CD34^+^ cells in the peripheral blood in comparison to non-smokers. This finding is consistent with the findings of Cohen et al. [14], who confirmed the CD34^+^ cell count dependence on smoking in healthy subjects, and Michaud et al. [15], who observed a reduced number and functionality of endothelial progenitor cells in healthy smokers. Similarly, Vasa et al. [16] confirmed the same effect of smoking on the number of CD34^+^ cells in patients with coronary artery disease. In our study, there was a borderline statistically significant decrease in number of CD34^+^ cells in active smokers in comparison to past smokers. This finding suggests that quitting smoking has a beneficial effect on the number of CD34^+^ cells, which, however, does not reach the same high level as in non-smokers. Cigarette smoke contains more than 4000 known compounds, including a large amount of free radicals and pro-oxidative factors. Oxidative stress is associated with reduced bioavailability of nitric oxide and thus impaired endothelial function [17]. It has been demonstrated that oxidative stress can reduce the mobilisation of endothelial progenitor cells from the bone marrow, affect their longevity, and impair their functional capacity [15].

A similar pathophysiological mechanism—oxidative stress and reduced bioavailability of nitric oxide in a hyperglycaemic state—could also be attributed to the depletion of the number and function of CD34^+^ cells in patients with diabetes mellitus [18,19]. On a molecular level, there is an excessive DNA methylation in CD34^+^ cells in a hyperglycaemic environment [20]. A linear correlation between the decrease in the number and functionality of circulating progenitor cells (including CD34^+^ cells) and the severity of diabetes and its vascular complications has been demonstrated [21]. However, with the improvement in cardiovascular health and increased physical activity in patients with diabetes, their number of CD34^+^ cells increased [22]. As shown in Figure 1, our PAD patients with diabetes had a borderline significant decrease in the number of CD34^+^ cells in peripheral blood; the lack of significance could be due to the small number of study participants.

Interestingly, the count of CD34^+^ cells in peripheral blood was lower in patients with longer claudication distances in our study. As previously proven [23], ischaemia is a very powerful stimulus for the migration of CD34^+^ cells from the bone marrow into peripheral blood, where they subsequently participate in angiogenesis, either by self-incorporating into the vascular wall and differentiating into mature endothelial cells or by producing and excreting angiogenic factors [24]. Therefore, the higher degree of ischaemia present in individuals with shorter claudication distances might present a greater stimulus for the mobilisation of CD34^+^ cells. Interestingly, the two objective indicators of PAD severity—ABI and toe pressure—did not demonstrate a correlation with CD34^+^ cell count, nor were they in any statistically significant correlation to claudication distance. This might be explained by the low number of study participants with rather similar characteristics.

We failed to confirm any correlation between the number of CD34^+^ cells and markers reflecting inflammatory activity, although inflammation is a recognised mechanism of PAD progression [25]. The proinflammatory cytokine, TNFα, inhibits the bone marrow and negatively affects the endothelial function [26], so it can be expected to be raised in patients with a low CD34^+^ count; however, we did not find a statistically significant correlation between the levels of TNFα and the number of CD34^+^ cells in our participants, nor did we find any studies researching similar relationships in patients with PAD. In patients with heart failure, Ugovšek et al. [27] did show a negative correlation between the level of TNFα and the level of CD34^+^ cells in peripheral blood, which indicates the importance of further research in this area.

Our research has several limitations. The most important one is the small number of subjects included. It is possible that with a larger number of subjects, the associations between some of the observed variables and the number of CD34^+^ cells would prove to be statistically significant. However, at a given time of study enrolment, the number of patients who met the strict inclusion criteria was very small. The majority of patients did not enter the study because they were suitable for revascularisation, and among patients with chronic limb-threatening ischaemia and no treatment options, many were excluded due to the presence of tissue necrosis, which would affect the values of laboratory variables, especially the observed inflammatory cytokines. Another important drawback is the absence of a control group, but we must be aware that due to the nature of the disease, patients with PAD represent an older population with a number of comorbidities, so it is almost impossible to form a control group that is free of associated diseases that affect the number of CD34^+^ cells in the peripheral blood.

## 4. Materials and Methods

### 4.1. Patients 

Our cross-sectional study included 30 patients with advanced PAD in the form of either chronic limb-threatening ischaemia without necrosis or intermittent claudication with a claudication distance ≤ 400 m, treated at the Clinical Department of Vascular Diseases, University Medical Centre Ljubljana, Slovenia. Exclusion criteria were clinical or laboratory signs of acute or chronic inflammation, the presence of gangrene or ulcers, a history of cancer in the last 5 years, bone marrow disease, and inability to perform a walking test in standard conditions (velocity of at least 3.2 km/h and 12.5% incline). The inclusion algorithm is presented in Figure 3.

The study was approved by the National Medical Ethics Committee of the Republic of Slovenia (reference number: KME 0120-143/2018/7, date of approval: 12 June 2018) and conducted according to the guidelines of the Declaration of Helsinki. All the patients gave their informed consent for participation.

### 4.2. Clinical Examination

The following clinical characteristics of participants were obtained through a questionnaire and clinical examination: sex, age, BMI, ABI, toe pressure, presence of arterial hypertension, diabetes mellitus, dyslipidaemia, and smoking. In smokers, pack-years were calculated as a product of the number of packs smoked per day and the number of years smoking. Based on their smoking status, the participants were categorised into three groups—active, past (cigarette-free for at least 1 year as of the day of inclusion in the study), and non-smokers. A thorough clinical examination was performed and blood pressure was measured on the arm and on the ankle to calculate the ABI. The toe pressure was measured using the Atys Medical plethysmograph (Soucieu-en-Jarrest, France). All the participants performed a walking test on a treadmill according to a standard protocol. The maximal distance until ischaemic pain occurrence was registered as claudication distance. All three tests are commonly used for the assessment of PAD severity [28,29]

### 4.3. Laboratory Tests

The patients’ venous blood was sampled from the median cubital vein in the morning hours, after 12 h of fasting. All the laboratory tests were performed by the laboratory of the Clinical Institute of Clinical Chemistry and Biochemistry, University Medical Centre Ljubljana. Total cholesterol, high-density lipoprotein (HDL), triglycerides, and apolipoproteins A1 and B were determined in the fresh serum by standard colourimetric or immunologic assays on the automated biochemistry analyser Fusion 5.1 (Ortho-Clinical Diagnostics, Raritan, NJ, USA). The same biochemistry analyser was used to determine lipoprotein (a) with the Denka reagent (Randox, Crumlin, UK), which contains apo(a) isoform-insensitive antibodies, and therefore showed minimal apo(a) size-related bias. The Friedewald formula [30] was used to calculate low-density lipoprotein (LDL). The level of haemoglobin A1c was determined using haemoglobin capillary electrophoresis. The estimated glomerular filtration rate (eGFR) was calculated using the MDRD (modification of diet in renal disease) formula [31]. Measurements of TNFα, high sensitivity reactive protein C (hs-CRP), interleukins 6 (IL-6), 8 (IL-8), and 10 (IL-10), angiopoietin-2, vascular endothelial growth factor (VEGF), and vascular cell adhesion molecule 1 (VCAM-1) levels in serum were determined using the Luminex xMAP Technology utilising magnetic beads coupled with specific antibodies, which allowed multiplexing. All the analyses were performed according to the manufacturer’s instructions (R&D Systems, Abingdon, UK). For haematological examinations, 3 mL of blood was taken into a tube with EDTA (Vacutainer, Becton Dickinson, Plymouth, England). We performed the tests on the same day from the fresh whole blood. The number of erythrocytes, leukocytes, and platelets, and the concentration of haemoglobin were determined using the device for automatic haematology measurements Cobas Minos STEX (Roche, Basel, Switzerland), and in doing so, we followed the standard procedures and instructions of the analyser manufacturer.

### 4.4. Immunomagnetic Positive Selection of CD34^+^ Cells and Cell Counting

To determine the number of CD34^+^ cells, 3 mL of blood was collected in a tube with ethylenediaminetetraacetic acid (EDTA) (Vacutainer, Becton Dickinson, Plymouth, England). Stem cells were isolated from peripheral blood using an Amicus cell separator (Baxter Healthcare, Deerfield, IL, USA). The magnetic cell separator, Isolex 300i (Nexell Therapeutics INC, Irvine, CA, USA), was used for the immunomagnetic positive selection of CD34^+^ cells. In the closed system, the collected cells were washed to remove the platelets, sensitised with mouse monoclonal anti-CD34 antibodies, and then incubated with immunomagnetic beads coated with polyclonal sheep anti-mouse antibodies (Dynabeads, Dynal AS, Oslo, Norway). The bead/CD34^+^ cell rosettes were separated in the magnetic field from other cells, and the CD34^+^ cells were released from the Dynabeads using an octapeptide with an affinity for anti-CD34 antibodies. The cells were then counted using a flow cytometer (Beckman Coulter Inc., Brea, CA, USA).

### 4.5. Statistical Analysis 

Statistical analysis was performed using SPSS (IBM SPSS Statistics 28.0.1.1., Armonk, NY, USA). Categorical variables were expressed as the number of units (N) and proportion (%). In numeric variables, the distribution pattern was tested by the Kolmogorov–Smirnov test. Normally distributed variables were expressed as the mean and standard deviation (SD), and asymmetrically distributed variables as the median and interquartile range. Differences between groups were compared with Pearson’s χ^2^ test or Fisher’s exact test for categorical variables, while independent continuous variables were compared with the Mann–Whitney U test, if asymmetrically distributed, and independent samples *t*-test if normally distributed. Differences among the three groups regarding smoking status were tested employing an analysis of variance (ANOVA) with Bonferroni correction for multiple testing. To assess the associations between the observed variables, Pearson’s correlation coefficient was used. A *p*-value of less than 0.05 was considered statistically significant.

## 5. Conclusions

In patients with advanced PAD and without the possibility of revascularisation, treatment with CD34^+^ cells poses a very promising method. However, one of the main factors that determine the success of the treatment is the number and functionality of CD34^+^ cells available for application. In our research, we demonstrated that smokers and patients with type II diabetes have a significantly reduced number of CD34^+^ cells in peripheral blood in basal conditions. Since the number of CD34^+^ cells before and after stimulation with G-CSF is related, such patients might be poor candidates for this type of treatment. Our finding that the number of CD34^+^ cells is inversely proportional to claudication distance indirectly indicates that ischaemia induced through interval training could increase the number of CD34^+^ cells.

## Figures and Tables

**Figure 1 ijms-24-15346-f001:**
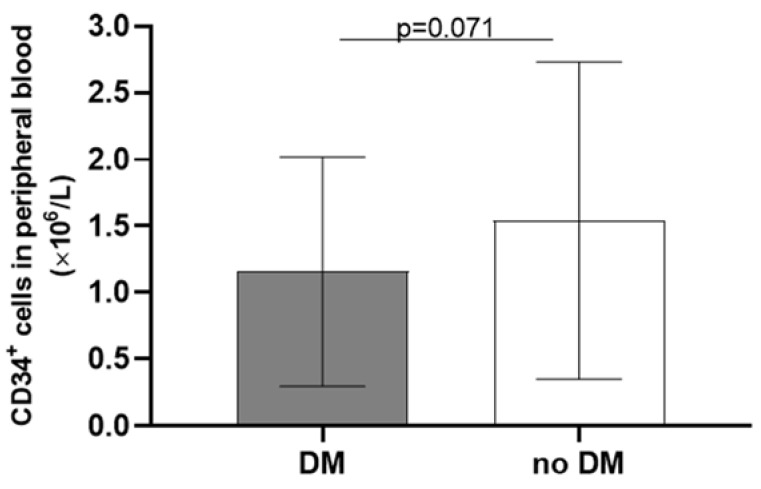
Impact of diabetes mellitus on the number of CD34^+^ cells in the peripheral blood of patients with peripheral artery disease.

**Figure 2 ijms-24-15346-f002:**
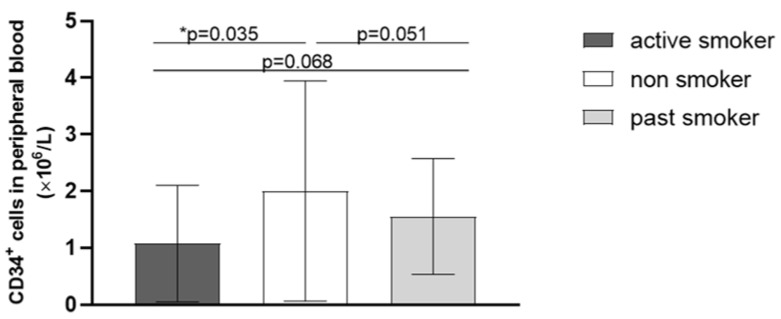
Impact of smoking on the number of CD34^+^ cells in the peripheral blood of patients with peripheral artery disease, * statistical significance at a level of *p* < 0.05.

**Figure 3 ijms-24-15346-f003:**
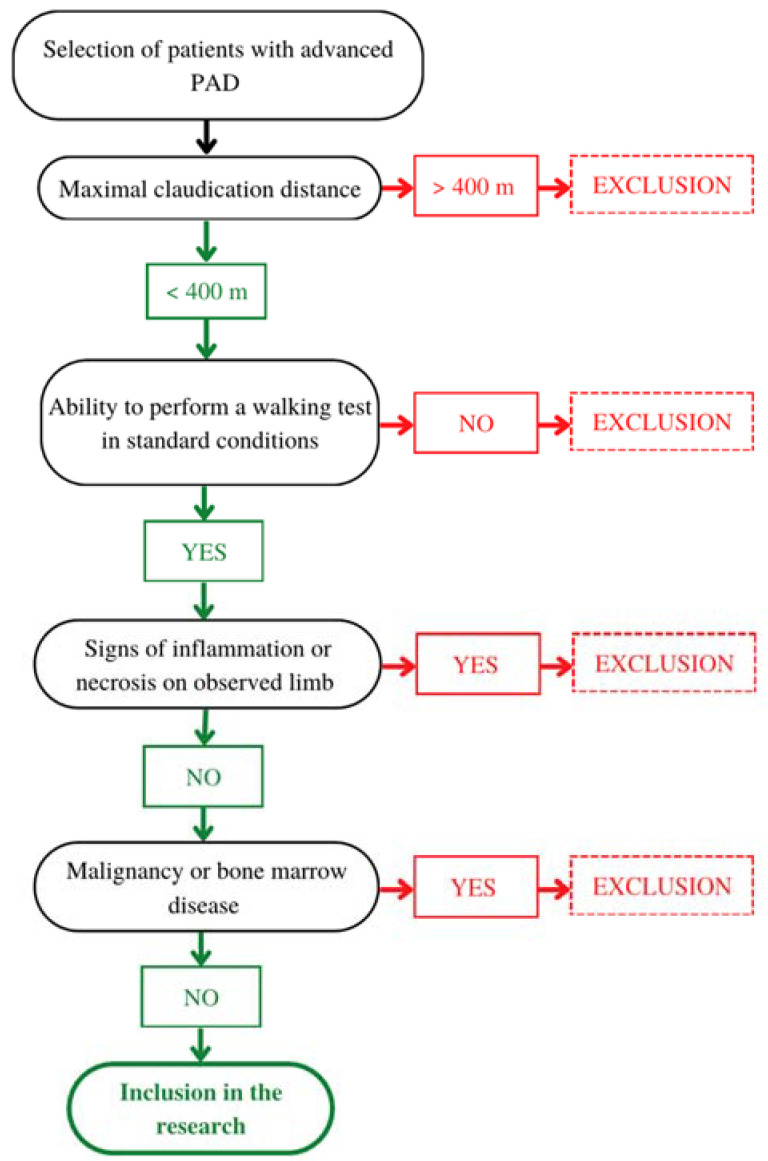
Inclusion process algorithm.

**Table 1 ijms-24-15346-t001:** Clinical characteristics of study participants.

Clinical Characteristics	
Male sex	23 (76.7)
Age (years)	68 ± 9
Body mass (kg)	79.7 ± 16.5
BMI (kg/m^2^)	26.7 ± 4.7
>30 kg/m^2^	9 (30)
Diabetes type II	12 (40.0)
Chronic kidney disease (eGFR < 60 mL/min/1.73 m^2^)	6 (20.0)
Dyslipidaemia	19 (63.3)
Arterial hypertension	26 (86.7)
Smokers; active/past	12 (40.0)/16 (53.3)
Chronic limb-threatening ischaemia	2 (6.6)
Previous revascularisation	11 (36.7)

Data are presented as the number and proportion of subjects or as the mean ± standard deviation. The estimated glomerular filtration rate (eGFR) was calculated by the MDRD (modification for diet and renal disease). BMI—body mass index.

**Table 2 ijms-24-15346-t002:** Biochemical parameters of study participants.

Laboratory Values	
Haemoglobin (mg/L)	99 ± 85
Leukocytes (×10^9^/L)	9.6 ± 2.6
Thrombocytes (×10^9^/L)	241 ± 75
CD34^+^ cells (×10^6^/L)	0.42 ± 0.72
Glucose (mmol/L)	7.26 ± 2.21
HbA1c (%)	6.10 ± 0.72
Urea (mmol/L)	9.13 ± 3.02
Creatinine (μmol/L)	98.03 ± 33.91
eGFR (mL/min/1.73 m^2^)	67.0 ± 21.7
Cholesterol (mmol/L)	4.37 ± 0.46
HDL (mmol/L)	1.21 ± 0.45
LDL (mmol/L)	2.35 ± 0.20
Triglycerides (mmol/L)	1.84 (1.15–2.37)
Apolipoprotein A1 (g/L)	1.50 ± 0.43
Apolipoprotein B (g/L)	0.88 ± 0.06
Lipoprotein (a) (mg/L)	105.93 ± 319.22
Angiopoietin 2 (ng/L)	2.03 ± 1.07
VEGF (ng/L)	0.15 ± 0.09
VCAM-1 (ng/mL)	721.10 ± 242.66
TNFα (ng/L)	37.92 ± 52.04
IL-6 (ng/L)	1.26 ± 1.26
IL-8 (ng/L)	59.35 ± 37.26

Data are presented as the mean ± standard deviation, except for triglycerides, which are presented as the median and interquartile range. HbA1c—glycated haemoglobin; eGFR—estimated glomerular filtration rate; HDL—high-density lipoprotein; LDL—low-density lipoprotein; VEGF—vascular endothelial growth factor; VCAM-1—vascular cell-adhesion molecule-1; TNFα—tumour necrosis factor α; IL-6—interleukin 6; IL-8—interleukin 8.

**Table 3 ijms-24-15346-t003:** Haemodynamic tests of participants’ peripheral artery disease severity.

Haemodynamic Tests	
Ankle–Brachial Index	0.51 ± 0.16
<0.4	7 (23.3)
0.41–0.9	21 (70.0)
0.91–1.4	2 (6.7)
Toe pressure (mm Hg)	64 ± 29
<30	3 (10)
30–70	15 (50)
>70	12 (40)

Data are presented as number and proportion of subjects or as mean ± standard deviation.

**Table 4 ijms-24-15346-t004:** Correlation between the number of CD34^+^ cells and some clinical characteristics of study participants.

Clinical Characteristics	Correlation
Age	r = 0.002; *p* = 0.993
BMI	r = 0.112; *p* = 0.557
Smoking (pack-years)	r = −0.300; *p* = 0.107

BMI—body mass index.

**Table 5 ijms-24-15346-t005:** Correlations between the number of CD34^+^ cells and markers of peripheral artery disease severity.

Parameter	Correlation
ABI	r = 0.165; *p* = 0.392
Toe pressure	r = −0.102; *p* = 0.598
Claudication distance	**r = −0.403; *p* = 0.046**

ABI—ankle–brachial index. Bold values indicate statistical significance at a level of *p* < 0.05.

## Data Availability

The data presented in this study are available on request from the corresponding author. The data are not publicly available due to protection of the privacy of personal data.

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
