# Peer review of "Smoking and Diabetes Attenuate Number of CD34+ Haematopoietic Stem Cells in Peripheral Blood of Patients with Advanced Peripheral Artery Disease"

_ijms, 2023, doi:10.3390/ijms242015346_

Round 1
Reviewer 1 Report
The author firstly examined the connection between the CD34+ cells count and the functional markers and risk factors for atherosclerosis in patients with the advanced PAD, and found that smokers and patients with type II diabetes have a significantly reduced number of CD34+ cells in basal conditions. The manuscript is well-recognized but there are only two issues associated with publication:
1) Figure 1, the caption is not completed, please replenish it.
2) The author has demonstrated the correlation of CD34+ cells count and smoking and diabetes type II, it’s better to explain why?
Moderate editing of English language required
Reviewer 2 Report
Reviewer comments and suggestions
The authors in this study explored the number of CD34+ hematopoietic cells in the peripheral blood of Peripheral artery disease (PAD) patients associated with patients' functional as well as atherogenic factors. The authors selected 30 patients with advanced PAD, recorded their performance in a walking test in standard conditions, and sampled their blood for further analysis with an emphasis on CD34+ cells selection and counting. The results included that the number of CD34+ cells differed between patients with and without type II diabetes (p=0.071) and between active smokers, past smokers, and non-smokers (p=0.035; p=0.068, p=0.051, respectively), with both smoking and presence of diabetes showed a negative effect on the number of CD34+ cells.
Overall, the manuscript was well written. However, a few major concerns/comments needed to be explained/modified.
- Line 44-45 Please add references for these studies.
- Line 60 (how is the amount) Is this sentence appropriate, please modify it with a good one
- Did the authors check the power analysis, yes they included it as well?
- Section 2.3 They can discuss the functional markers in a comprehensive way
- Line 104-105 is there any possible reason that can be also mentioned here
- Comments for Figure 1 figures show non significant, how the authors explain it
- Line 115 I think the authors could mention diabetes as well.. rather than advanced PAD, because they have included it
- Line 136-141 How can the authors defend their studies with the previous one
- The authors could present a well set up mechanism even though they did not find a significant result with the help of a diagram if possible
- Line 281 GSF This was the first time used, better to explain in the introduction or discussion section
- A few references need to be modified, please check it again
